# Factors associated with retention of mother-baby pairs in the elimination of mother-to-child transmission of HIV program in Kaberamaido district: A longitudinal analysis

**James Daniel Odongo**[1,2]*, **Ronald Opito**[3], **Benon Wanume**[1], **Denis Bwayo**[4], **David Mukunya**[1], **Samuel Okware**[5], **Joseph K. B. Matovu**[1,6]

**1** Department of Public Health, Busitema University, Mbale, Uganda, **2** Department of Health, Kaberamaido District Local Government, Kaberamaido, Uganda, **3** Department of Public Health, School of Health Sciences, Soroti University, Soroti, Uganda, **4** Department of Internal Medicine, Busitema University, Mbale, Uganda, **5** Uganda National Health Research Organization, Entebbe, Uganda, **6** Department of Disease Control and Environmental Health, Makerere University School of Public Health, Kampala, Uganda

* jamesdodongo@yahoo.com

**Data Availability Statement:** All relevant data are within the paper and its Supporting Information files.

## Abstract

### Background

Retention along the elimination of Mother to Child Transmission (eMTCT) cascade in Uganda remains poor as only 62.7%-69.5% are followed up to 18months. The objective of this study was to determine the rates of retention of mother-baby pairs at two levels of the eMTCT cascade (12 and 18 months) and associated factors.

### Methods

This was a longitudinal analysis of 368 mother-baby pairs who were enrolled into the eMTCT program in Kaberamaido district from January 2013 to December 2018. Data was extracted from early infant diagnosis (EID) and mothers' ART registers, entered into Microsoft Excel and then exported to Stata statistical software package version 14.0 for management and analysis. Descriptive statistics such as mean and frequencies were computed at univariate level. At the bivariate level, Cox proportional hazard regression was performed to assess the level of association between the primary outcome and each independent variable, while Cox proportional hazard regression model was built at multivariate level to determine the factors independently associated with retention of mother-baby pairs in the eMTCT program.

### Results

Of the 368 mothers enrolled into the study, their average age was 29.7years (SD = 6.6). Nearly two-thirds of the mothers were married/cohabiting, (n = 232, 63.0%). The 368 mother baby pairs were observed for a total time of 6340 person months, with majority, 349 (94.8%, 95%CI = 92.0–96.7) still active in eMTCT care, while 19(5.2%, 95%CI = 3.3–8.0) were lost to follow up at 12months. At 18 months, 323 (87.8%, 95%CI = 84.0–90.8) were active in eMTCT program while 45(12.2%, 95 CI = 9.2–16.0) were lost to follow up. At bivariate level,

**Funding:** The authors received no specific funding for this work.

**Competing interests:** The authors have declared that no competing interests exist.

marital status, health facility level of enrolment, mothers' ART treatment supporter, and mothers' ART enrolment time were significantly associated with survival/lost to follow up (LTFU) of mother-baby pairs along the eMTCT cascade. At multivariable level, the mothers' time of ART initiation was significantly associated with survival/lost to follow up (LTFU) of mother-baby pairs at along the eMTCT cascade, with mothers-baby pairs who were initiated during the antenatal/post-natal periods having higher hazards of LTFU compared to those who initiated ART before Antenatal period (before pregnancy), aHR = 4.37(95%CI, 1.62–11.76, P = 0.003). Mother-baby pairs who were enrolled into the eMTCT program after the implementation of HIV test and treat policy (year 2017 and 2018) had higher hazards of LTFU as compared to those enrolled before the implementation of test and treat policy in Uganda (year 2013–2016), aHR = 2.22(95% CI, 1.15–4.30, P = 0.017). All the other factors had no significant association with lost to follow up and cascade completion at multivariate level.

## Conclusion

There was high level of retention of mother-baby pairs in the eMTCT program in Kabera-maido at 12 months, but it was suboptimal at 18months. ART initiation during the antenatal and/or post-natal period was significantly associated with suboptimal retention of mother-baby pairs along the eMTCT cascade.

## Introduction

In Uganda, vertical transmission of HIV is the major source of HIV infections in children, contributing 14% of new infections annually. Although vertical transmission of HIV has declined to 5.8% over the past 5 years, it remains higher than the national set target of less than 2% [1]. In Uganda, there has been a dramatic reduction in new vertical infections from 25,000 in the year 2000 to about 3486 in 2015 after roll out of Option B+ since 2013 [2]. However, in 2021, there were 53000 new HIV infections, of which 5700 were Mother To Child Transmission, which is 10.7% of the total new infections [1].

Without treatment, the risk of HIV transmission from mother-to child ranges from 15% to 45%. This risk can be reduced to less than 5% in mothers on antiretroviral treatment (ART) if the mother utilizes all the services in the eMTCT package [3]. Packages in the eMTCT program include Prevention of HIV infection among women of child bearing age, preventing unintended pregnancies among women living with HIV, preventing HIV transmission from woman living with HIV to her baby through option B+ and provision of effective treatment, care and support to HIV+ mothers, their children and families. Although HIV testing in Antenatal Care (ANC) is almost universal in Uganda, retention along the elimination of Mother to Child Transmission (eMTCT) cascade remains poor as only 56.3% are tested for final EID test at 18 months [3], 31% have the second Polymerase Chain Reaction (PCR) and 44% have the first PCR test, indicating that 13% of HIV Exposed Infants (HEIs) dropout between first PCR and second PCR tests before the final rapid test is done at 18months [4]. As a result, many HIV infant infections are now occurring during the postnatal period rather than pregnancy or labour [5]. The cumulative risk of continued breast-feeding for two years without any intervention is 20–45% [6].

The risk of transmission ranges from 5% to 10% during pregnancy, 10% to 15% during labour/delivery, and 5% to 20% through breastfeeding without prolongation to 2years. Among

the key interventions developed and implemented to address the eMTCT challenges are the integration of eMTCT and HEI services into the maternal, newborn, child and adolescent health (MNCAH) service delivery platform. This is aimed at providing a one stop point of delivery of these services [2].

Retention levels of mother-baby pairs are very low, ranging between 62.7%-69.5% [7, 8]. Strategies that have been documented to improve retention of mother–baby pairs in the eMTCT program include addressing barriers affecting eMTCT program. The barriers identified are, non-disclosure, lack of psychosocial support, distance to health facilities, HIV-related stigma and discrimination, poverty, under staffing and long waiting time among others [9, 10].

HIV-related stigma and discrimination has been found to negatively influence a pregnant woman's decision to enrol in and utilize the eMTCT program and subsequently interrupt adherence to treatment and retention in care [11]. Being unaware of one's HIV status can act as a barrier to utilization of the eMTCT services at various levels of the eMTCT cascade. The point at which women are tested for HIV can also impact on their journey through the eMTCT cascade, should they test positive. For instance, a study of HIV sero-positive pregnant women conducted by Dionne-odom in Cameroon, Cote d'Ivoire, South Africa, and Zambia found that women who were diagnosed with HIV before their pregnancy were more likely to adhere to the eMTCT program than women who tested positive during pregnancy [12].

Further, some studies have shown that despite mothers having high levels of HIV, MTCT and eMTCT awareness, there can be still low acceptability of eMTCT services. For instance, a study from south west Nigeria found that, despite 99.8% of pregnant women being aware of HIV and with very high knowledge of MTCT (92%) and eMTCT (91%), 71% had negative views towards eMTCT and did not use the HIV services which was meant to be part of the eMTCT interventions [13]. Another study which looked at the postnatal knowledge of mothers showed that eMTCT knowledge and younger age of the mother were associated with pregnancy desire, and among HIV negative women HIV disclosure to the partner, younger age of the mother and having a lower number of children were associated with pregnancy desire and the subsequent adaption of the eMTCT Program [14]. The objective of our study was to determine levels of retention at two levels of the eMTCT cascade (at 12 and 18 months) and the factors associated with retention of mother-baby pairs at those levels.

## Materials and methods

### Study area

This study was conducted in five health facilities in Kaberamaido district that are accredited to offer ART and eMTCT services. These included Kaberamaido Hospital, Kobulubulu HCIII, Ochero HCIII, Alwa HCIII, and Kaberamaido Catholic Mission HCIII. The eMTCT services offered in these health facilities include provision of HIV testing services for all pregnant women during antenatal care (ANC), provision of antiretroviral therapy (ART) for all HIV positive pregnant women during ANC, labour and post-natal period, 1st PCR test for exposed infants at 6weeks and linkage to ART for the positive infant, 2nd PCR test for exposed infant at 12months and linkage to ART for the positive infant, and rapid HIV testing at 18months and exit of the mother-baby pairs if the infant is negative at this point [2].

Provision of these services is supported by The AIDS Support Organization (TASO) Uganda Limited through a health system strengthening grant from United States Centre for Disease control and prevention (US-CDC). TASO has been implementing 5-year health systems strengthening project in the Soroti region with the purpose to "Achieve Epidemic Control through the attainment of 95-95-95 UNAIDS targets by 2020 and strengthening health

systems in the Soroti Region in the Republic of Uganda under the President's Emergency Plan for AIDS Relief (PEPFAR)." [15]. One of the activities conducted in this grant is strengthening the capacity of the health facility staff in Kaberamaido district to provide a complete and high quality eMTCT program to the mother-baby pairs, which include early identification of HIV positive pregnant mothers and initiation of ART, provision of nevirapine prophylaxis on time, provision of EID services and follow up of the mother-baby pairs who have missed their appointments, upto the point of discharge from the program at 18months.

Kaberamaido District is one of the districts in Teso sub-region located in Eastern Uganda, it is bordered by districts of Amolatar in southwest, Dokolo in the North and northwest, Kalaki in the East, and Lake Kyoga in the south and southeast, sharing the shores of this lake with the districts of the Busoga region. The district headquarters is situated within Kaberamaido Town Council; approximately 434 km to the east of Kampala City; the capital of the Republic of Uganda. By 2014, the district had an estimated total population of 215,026 (105,152 in Kaberamaido County, the present day Kaberamaido district) [16].

The adult population HIV prevalence currently stands at 4.0% compared to the previous 7.8% in 2006 [1]. PMTCT program began in 2013 after the government of Uganda adopted option B+ in 2012 and this is being implemented in level three facilities and above. According to the district HIV/AIDS data (District health information systems-DHIS2, unpublished), about 914 women were enrolled into chronic HIV care with more than half of these still within the reproductive age. Kaberamaido is one of the districts in the eastern region with the lowest mother-baby-pair retention rates at 60% at 18 months as per the national reports, this means that 40% of mother-baby pairs are lost to follow up which is way below the recommended 95% retention and <5% loss to follow up by the UNAIDS and MoH.

## Study design

This was a longitudinal analysis using secondary data routinely collected for mother-baby pair management at the health facilities. This method was chosen because these data were readily available, and the mother-baby pairs could be followed in the registers from the point of registration up to the point of exit or getting lost. A data abstraction tool was designed based on the early infant diagnosis (EID) register and maternal ART register to capture basic maternal demographic characteristics and progression along the eMTCT cascade. Data were abstracted for the eMTCT mother-baby pairs enrolled into the program from 1st January 2013 to 31st December 2018 and followed for at least 18 months for progression along the cascade.

## Study population

The study population were all HIV-positive mother-baby pairs enrolled in the eMTCT program from 1st January 2013 to 31st December 2018 in Kaberamaido District. The mother-baby pairs included in the study were those enrolled into the eMTCT program from 1st January 2013 to 31st December 2018 in the 5 health facilities accredited to offer ART. Mother-baby pairs enrolled into the eMTCT program from 1st January 2013 to 31st December 2018 in Kaberamaido, registered in both early infant diagnosis (EID) and ART registers in Kaberamaido and with complete information for both the mothers and the HIV exposed infants (HEIs) were included in this study. All the mother baby pairs were followed up for a maximum period of 18months from the date of birth of the infant, and all those who were still active at 18 months were censored. Censorship was also done for clients who were declared lost to follow up before reaching 18months. On the other hand, clients with incomplete/missing information, clients transferred from other health facilities and clients not in the EID/ART register were excluded from the study.

## Sample size determination

The sample size was determined using the Cochran Equation, that is; n = $Z^2pq/e^2$, where *n* is the sample size, *Z* is the value from the table of probabilities of the standard normal distribution for the desired confidence level (e.g., Z = 1.96 for 95% confidence), *p* is the probability of the event occurring, and *q* is the probability of the event not occurring, *e* is the level of precision which is 5%. Considering the eMTCT retention rate at 18months in Gulu, Northern Uganda which was reported at 69.5% [7] our *p* therefore is 69.5% = 0.695, and q = 1-p = 0.305, resulting into a sample size 326. Adjusting 5% for incomplete data, sample size determined was 342. When we reviewed records, we found 457 mother-baby pairs enrolled in the eMTCT program, of which 368 had complete data and decided to collect and analyse data on all the enrolled mother-baby pairs who were eligible to be included in the analysis. (Fig 1 shows the flow chart of enrolment and survival along the eMTCT cascade).

## Data abstraction procedures

Data collected for the eMTCT program was extracted from the primary data tools (EID register, baby's EID chart/card & ART register/care card) using the study data abstraction tool. Data from all mother-baby pairs enrolled from 1[st] January 2013 to 31[st] December 2018 were extracted and those with complete information were considered for analysis. Data on socio-demographic characteristics such as age, marital status, education level, and treatment supporter were extracted from the registers. Other independent variables (Employment, place of residence, distance to the health facility estimated using google map, mothers' ART start time, ANC attendance by looking at the ANC number, mode of delivery, sero-status disclosure were obtained from the mother's ART register, age of the baby and PCR test results, infant ARV prophylaxis status) and the outcome variables of interest which was cascade completion at 18 months post-delivery were extracted from MoH EID and ART registers. Pretesting and modification of the developed data abstraction tool was done prior to data extraction to check its suitability and ease of use to extract the required information. Two research assistants were trained to extract data from EID and ART registers in the health facilities implementing the eMTCT Program.

## Measurement of variables

The main outcome variables were the rate of mother-baby pairs retention at 12 and 18 months measured by determining the proportions of mother-baby pairs who were alive and active in care at 12 and 18 months from the date of birth in the eMTCT Program. A mother-baby pair was considered active if the pair had a future appointment date or the appointment date was within the last 30 days (missed drugs for not more than 30 days). The mother-baby pair was considered retained if both were alive and active in care at 12 months and 18 months after delivery. In case any of the members of the pair (either the mother or baby) was lost at any level of the cascade before 18 months, that pair was considered as lost to follow up (LTFU) and the rate was equally obtained at 12 and 18months. The independent variables included age of the mother considered as a continuous variable, HIV disclosure status (disclosed or not disclosed), point of ART initiation which was before pregnancy/ANC, during pregnancy/ANC or during lactation (Post ANC), Viral load of the mother (suppressed = Viral load<1000copies/μl or unsuppressed = viral load≥1000copies/μl) which was considered as the latest viral load done during the period of stay in the eMTCT program, education level of the mother, marital status of mother, treatment supporter of mother, year of enrolment into the eMTCT program, ANC attendance, time of initiation of nevirapine prophylaxis for the baby categorized as within 72 hours or beyond 72 hours, place of delivery which was either home or health facility, mode of delivery and age at first and second PCR test.

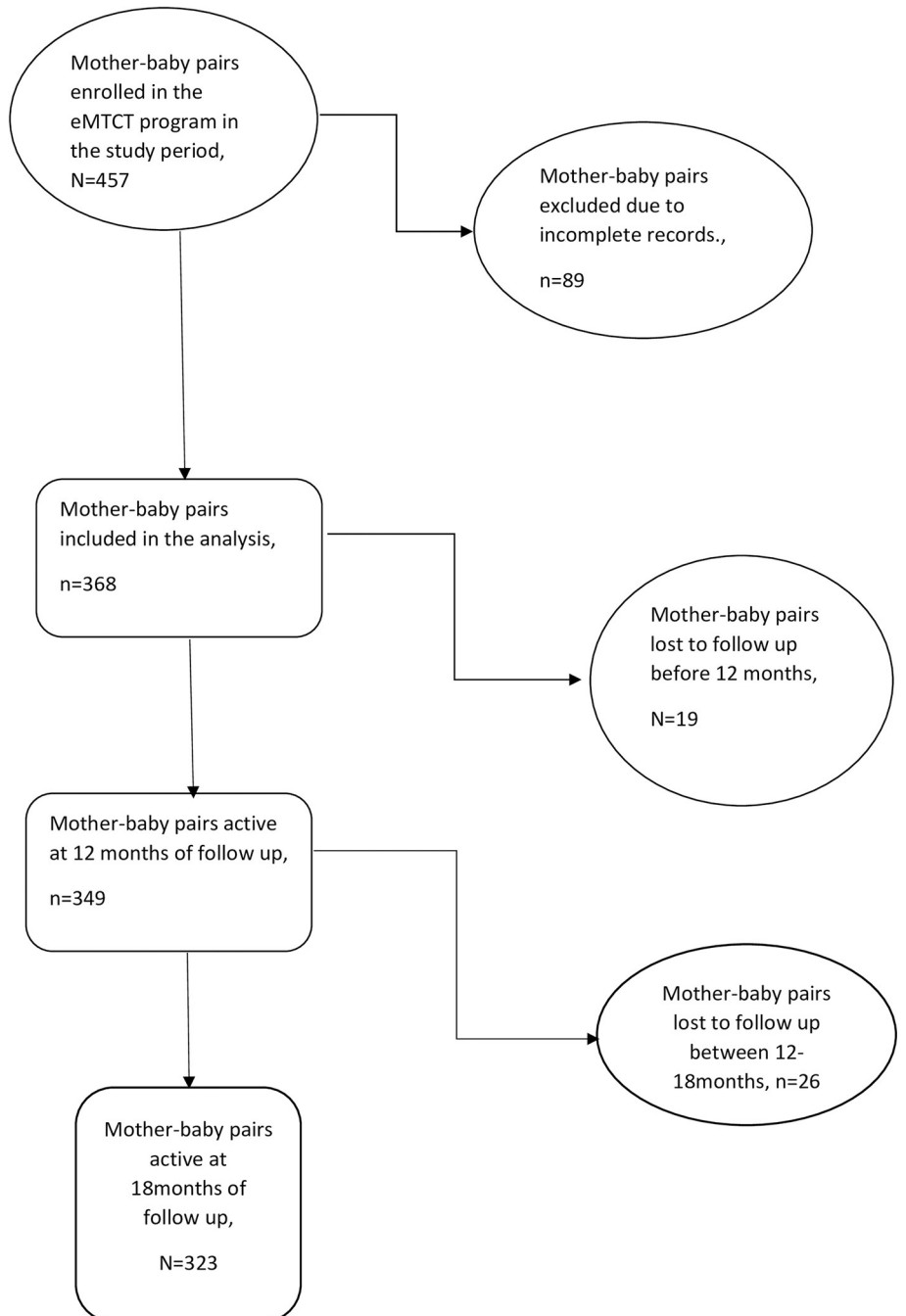

**Fig 1. A flow chart showing enrolment and survival along the eMTCT cascade.**

## Data analysis

Data were entered into Microsoft excel, cleaned for missing information, and then exported to Stata version 14.0 statistical software for further management and analysis. A Kaplan-Meier survival curve was generated to show trends in mother-baby pair survival (retention) along the eMTCT cascade. Quality assurance and validity of data were enhanced through training of research assistants on standardized data extraction procedures and strict supervision of data

extraction and entry. To ensure further consistency and accuracy of the data, double data entry was done. Descriptive statistics including mean, standard deviations, proportions and frequencies were computed at univariable level. At bivariable level, Cox proportional hazards regression was performed to assess the association (hazards) of lost to follow up against each independent variable along the eMTCT cascade (second PCR test and 18 months' cascade completion). Factors with a $p<0.25$ at bivariate level were entered into the final multivariable analysis level along with other factors of biological/clinical plausibility and potential confounders. A sub-group analysis was conducted to assess for factors associated with retention at the two levels of the eMTCT cascade (at second PCR test and at 18 months at cascade completion). We conducted a Cox proportional hazards model regression to assess the predictors of lost to follow up of mother-baby pairs along the eMTCT cascade. All factors with a $p<0.05$ were statistically significant and thus predictors of lost to follow up in the eMTCT Program.

## Ethical considerations

This study protocol was approved by the Mbale Regional Referral Hospital Research and Ethics Committee (MRRH-REC–REG No: UG-REC-011). We utilized Kaberamaido district health facility data of clients who enrolled for the eMTCT program from 1st January 2013 to 31st December 2018 and as such permission to utilize the health facility-based data and conduct the study in the district was sought and gotten from the Chief Administrative Officer (CAO) and District Health Officer (DHO) Kaberamaido district. The clearance and authorization letters from the MRRH-REC, local authorities were presented to the health facility incharges by the research assistants for them to be allowed to collect data.

## Results

### Socio-demographic and clinical characteristics of study participants

Table 1 shows the sociodemographic and clinical characteristics of 368 mother-baby pairs enrolled into the study. The mothers were on average aged 29.7years (SD = 6.6). Nearly two-thirds of the mothers were married/cohabiting, (n = 232, 63.0%). Majority of the mothers, 93.2% (n = 343) attended ANC at least once before delivery and most (n = 320, 87.0%) of the deliveries were in a health facility. Most of the mothers, 63.3% (n = 233) had their husbands as their treatment supporters, more than half of mother-baby pairs (n = 188, 51.1%) were enrolled into ART care through OPD, while the remaining 48.9% (n = 180) were enrolled through the eMTCT program. The mean infant age of conducting 1st PCR test was 9.1 (SD = 6.6) weeks, with a median age of 7 (IQR; 6–9) weeks and majority of infants received 1st PCR after 8weeks of birth, 70.6%(n = 260).

### Retention of mother-baby pairs along the eMTCT cascade in Kaberamaido district

The 368 mother baby pairs were observed for a total time of 6340 person months, with 349 (94.8%, 95%CI = 92.0–96.7) still active in PMTCT care, while 19(5.2%, 95%CI = 3.3–8.0) were lost to follow up at 12months. At 18 months, 323 (87.8%, 95%CI = 84.0–90.8) were active in PMTCT program while 45(12.2%, 95 CI = 9.2–16.0) were lost to follow up (Fig 2). The average time spent in the program by mother-baby pairs who were lost to follow up was 12.0(SD = 5.3) months.

### Factors associated with mother-baby pair survival/lost to follow up (LTFU) along the eMTCT cascade

Table 2 shows Factors associated with mother-baby pairs survival/lost to follow upalong the eMTCT cascade.

**Table 1. Socio-demographic and clinical characteristics of study participants enrolled in the eMTCT program between 2013 and 2018 in Kaberamaido district.**

| Characteristics | Population, N = 368 | Proportion (%) |
|---|---|---|
| Age of the mother, mean (SD) 29.7(6.6) | | |
| **Age of the mother at enrolment** | | |
| <20 | 16 | 4.4 |
| 20–24 | 62 | 17.1 |
| 25–40 | 262 | 71.2 |
| >40 | 27 | 7.3 |
| **Marital status at enrolment** | | |
| Single* | 136 | 37.0 |
| Married** | 232 | 63.0 |
| **Health facility of enrolment** | | |
| HCIII | 191 | 51.9 |
| Hospital | 177 | 48.1 |
| Year of Enrolment | | |
| Year 2013 | 13 | 3.5 |
| Year 2014 | 50 | 13.6 |
| Year 2015 | 70 | 19.0 |
| Year 2016 | 94 | 25.5 |
| Year2017 | 64 | 17.4 |
| Year 2018 | 77 | 20.9 |
| **Distance to health facility of enrolment** | | |
| ≤5km | 187 | 50.8 |
| >5km | 181 | 49.2 |
| **Mothers' ART entry point** | | |
| Outpatient Department | 188 | 51.1 |
| eMTCT | 180 | 48.9 |
| **Mothers' treatment supporter** | | |
| Husband | 233 | 63.3 |
| Other Relatives | 135 | 36.7 |
| **ANC attendance of recent pregnancy** | | |
| No | 25 | 6.8 |
| Yes | 343 | 93.2 |
| **Place of birth of current baby** | | |
| Health facility | 320 | 87.0 |
| Home | 48 | 13.0 |
| **Viral load results available** | | |
| No | 136 | 37.0 |
| Yes | 232 | 63.0 |
| **Viral load status (N = 232)** | | |
| Non-suppressed | 29 | 12.5 |
| Suppressed | 203 | 87.5 |
| **Mothers' ART enrolment Time** | | |
| Pre-pregnancy | 161 | 43.7 |
| Antenatal | 189 | 51.4 |
| Post-natal | 18 | 4.9 |
| **Infant ARV prophylaxis** | | |
| NVP≤72hrs | 281 | 76.4 |

(*Continued*)

**Table 1.** (Continued)

| Characteristics | Population, N = 368 | Proportion (%) |
|---|---|---|
| NVP>72hrs/No ARVs | 87 | 23.6 |
| **Infant age at 1ˢᵗ PCR, median = 7 (IQR; 6–9)** | | |
| ≤8weeks | 260 | 70.6 |
| >8weeks | 108 | 29.4 |

\* = Single, never married, separated, widowed or divorced

\*\* = Married or cohabiting.

At bivariate level, marital status, health facility level of enrolment, mothers' ART treatment supporter, and mothers' ART enrolment time were significantly associated with lost to follow up of mother-baby pairs at 18-months.

At multivariate level, the mothers' time of ART initiation was significantly associated with lost to follow up of mother-baby pairs at along the eMTCT cascade, with mothers who were initiated during the antenatal/post-natal periods having higher hazards of lost to follow up of mother-baby pairs compared to those who initiated ART before Antenatal period (before pregnancy), aHR = 4.37(95%CI, 1.62–11.76, P = 0.003). Mother-baby pairs who were enrolled into the eMTCT program after the implementation of HIV test and treat policy (those enrolled in year 2017 and 2018) had higher hazards of LTFU as compared to those enrolled before the implementation of test and treat policy in Uganda (those enrolled between 2013–2016), aHR = 2.22(95% CI, 1.15–4.30, P = 0.017). All the other factors had no significant association with lost to follow up and cascade completion at multivariate level.

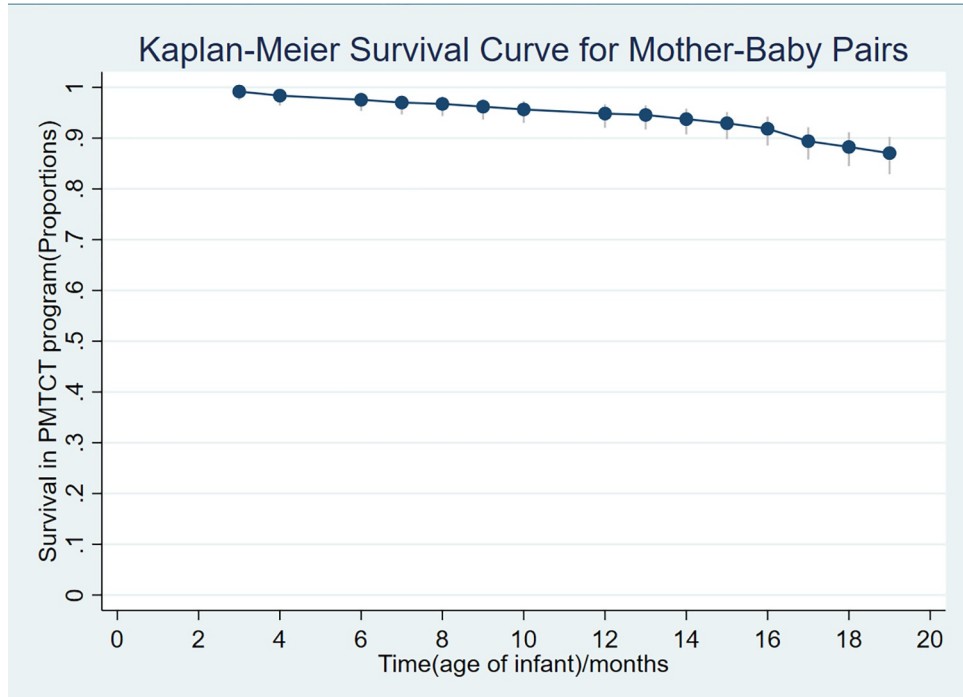

**Fig 2. Proportion of mother-baby pairs enrolled between 2013 and 2018 retained/survived along the levels of the eMTCT cascade in Kaberamaido District.**

**Table 2. Factors associated with survival/lost to follow up (LTFU) of mother baby pairs at along the eMTCT cascade.**

| Factor | LTFU before 18 months, n/N (%) | HR (95%CI) | aHR (95%CI) | P-Value |
|---|---|---|---|---|
| All | 45/368(12.2) | | | |
| Age of Mother/years | | 0.97(0.93–1.02) | 1.00(0.95–1.05) | 0.989 |
| Marital Status | | | | |
| Single* | 25/136(18.4) | 1.00 | 1.00 | |
| Married** | 20/232(8.6) | **0.45(0.25–0.80)** | 0.78(0.31–1.93) | 0.585 |
| Health Facility Level | | | | |
| HCIII | 14/191(7.3) | 1.00 | | |
| Hospital | 31/177(17.5) | **2.63(1.40–4.94)** | 2.08(0.97–4.46) | 0.059 |
| Year of Enrolment | | | | |
| Year 2013_2016 | 22/227(9.7) | 1.00 | 1 | |
| Year 2017_2018 | 23/141(16.3) | 1.74(0.97–3.12) | **2.22(1.15–4.30)** | **0.017** |
| Mothers ART Entry Point | | | | |
| OPD | 19/188(10.1) | 1.00 | 1.00 | |
| eMTCT | 26/180(14.4) | 1.46(0.81–2.64) | 0.78(0.39–1.56) | 0.484 |
| Mothers Treatment Supporter | | | | |
| Husband | 22/233(9.4) | 1.00 | 1.00 | |
| Other relatives | 23/135(17.0) | **1.90(1.06–3.41)** | 1.01(0.42–2.44) | 0.985 |
| ANC Attendance | | | | |
| No | 6/25(24.0) | 1.00 | 1.00 | |
| Yes | 39/343(11.4) | **0.41(0.17–0.97)** | 0.41(0.11–1.51) | 0.178 |
| Place of Delivery | | | | |
| H/Facility | 40/320(12.5) | 1.00 | 1.00 | |
| Home | 5/48(10.4) | 0.82(0.32–2.08) | 0.45(0.14–1.49) | 0.193 |
| Mothers ART Start Time | | | | |
| Pre-conception | 8/161(5.0) | 1.00 | 1.00 | |
| Ante/post-natal | 37/207(17.9) | **3.93(1.83–8.43)** | **4.37(1.62–11.76)** | **0.003** |
| Infant ARV Prophylaxis | | | | |
| NVP≤72hrs | 32/281(11.4) | 1.00 | 1.00 | |
| NVP>72hrs/No ARVs | 13/87(14.9) | 1.33(0.70–2.54) | 1.22(0.55–2.68) | 0.621 |
| Infant Age_CAT at 1st PCR. | | | | |
| ≤8weeks | 27/260(10.4) | 1.00 | 1.00 | |
| >8weeks | 18/108(16.7) | 1.64(0.90–2.97) | 0.55(0.28–1.08) | 0.083 |

* = Single, never married, separated, widowed, or divorced

** = Married or cohabiting, **Bold** = Significant with p<0.05, uHR = unadjusted hazard rastio, aHR = adjusted hazard ratio, all factors are adjusted for each other.

## Discussion

Our study found that retention of mother-baby pairs in care under the eMTCT program was 94.8% at 12 months and 87.8% at 18 months. The 12-month retention of 94.8% observed in this study shows a significant progress in eMTCT programming in the country as earlier studies like one in Tororo found overall 12-month retention under test and treat to be 78.7% [17]. Our study differs from previous ones in the East African region that have documented much lower and suboptimal retention in care, ranging from 60.2% in Gombe, 66.8% in Ntungamo, 81.5% in central Uganda (with intervention),74% in Rwanda and 89.7% in Kenya (after intervention) [9, 18–21]. This difference could be due to different support the district gets from the implementing partners such as TASO–Soroti Project that could have led to improved eMTCT services provision as well as improve knowledge and attitudes towards the eMTCT services

among sero-positive women in Kaberamaido District. With improved retention of mother-baby pairs in eMTCT program, it implies that virtual elimination of mother to child transmission of HIV in Uganda is very feasible and the government and implementing partners should put more efforts in ensuring early identification of HIV exposed infants and providing them with a comprehensive service.

The high retention observed in this study is an encouraging finding, that points towards likelihood of attainment of the UNAIDS 2030 target of at least 95% of clients on ART retained in care. This level of retention has potential to lead to virtual elimination of vertical transmission of HIV infection. This high level of retention could have been attained due to many factors such as health system strengthening support by implementing partners such as TASO Soroti Project, who recruited linkage facilitators, HIV counsellors, mentor mothers and HIV/AIDS expert clients to support clients at different levels of care. The support from these partners, has enabled the district to rollout the eMTCT services to lower health facilities hence reducing the distance and cost of transport to the point of care as well as improved differentiated service delivery. However, the 12.2% of mother-baby pairs who were not retained at 18months are most likely those who did not receive rapid test and not discharged. There is need to ensure that mothers are given adequate health education on the importance of completing the eMTCT cascade, so as to be discharged with an HIV negative child.

The study showed that the rate of LTFU was lower between initiation of ART and 12months of the eMTCT cascade, this is contrary to findings from studies by [9, 19] which found that LTFU was highest between ART initiation and the 6months visits, this may be explained by the improved knowledge among mothers on the importance of adhering to appointment dates in order to protect their babies from acquiring HIV infection, women empowerment especially on decision making on matters that affects their health. However, LTFU starts to increase as you move along the eMTCT cascade especially at 18months and above, probably due to the fact that as many babies are declared sero-negative, their mothers wean them off and do not see reasons to continue adhering to clinic appointments, since this also marks a period of transition from eMTCT program where services are offered in an integrated manner to the general ART clinic.

The factors found to be significantly associated with survival/LTFU along the cascade were the mothers' point of ART initiation and the year of enrolment into eMTCT program. Mothers who were initiated during the antenatal and/or post-natal period had higher hazards of LTFU along the eMTCT cascade before completion compared to those who initiated ART before pregnancy. The positive effects of these factors (diagnosed with HIV prior to pregnancy and early ART initiation) on eMTCT cascade completion could be explained by the fact that these mothers received several adherence counselling sessions prior to pregnancy packaged with information on the benefits of ART and importance of having an HIV free baby. This finding implies that retaining mothers in the eMTCT program and to achieve the desired goal of virtual elimination of MTCT, HIV programs have to put concerted efforts in early identification of HIV positive women so that they can be initiated on treatment prior to pregnancy and delivery. This needs more efforts in community education and sensitization as well as interventions geared towards early uptake of the eMTCT services.

Other factors such as young age of the mother which was found to be associated with poor eMTCT cascade completion in a systematic review study on retention in care during pregnancy and post-partum period in option B+ era in Africa [22], did not however have significant association with retention of mother-baby pairs at the two levels of the eMTCT Cascade in our study. This could have been due to presence of targeted services for adolescents and young women as well as rolling out of other differentiated service delivery models in the district.

## Study limitations and strengths

This study had several limitations and some strengths. The process of retrieving retrospective data collection was limited by some incomplete data in the registers. The formula used to calculate the sample size was inappropriate for determining the factors associated with retention in the eMTCT program. We ensured that all the clients who were enrolled in the selected facilities and had complete data were included to counter the sample size effect. There was limitation in determining true loss to follow up (misclassification of outcome) because some women who were captured as lost to follow up in the mother facilities might have had self-transfers and completed cascade in another facility, yet this information was not available. Since our study used secondary data collected retrospectively, some factors associated with Mother-Baby Pairs retention could have not been captured and hence were left out since studied factors were based on routinely recorded information. This study was not able to assess the health system factors associated with this improved retention. Further studies on the social and health system factors affecting retention of mother-baby pairs need to be conducted. The study was not able to assess the facilitators of improved retention in eMTCT program as we did not interview clients and these were not captured in the data sources used for the study. Some confounding factors such as stigma, political and socioeconomic factors, for instance, access to clinic, affordability, availability of services nearby; severity of illness were not considered in the analysis since it was secondary data, whereas they could have had an effect on retention of mother-baby pairs in eMTCT program. Despite the above-mentioned limitations, this study was unique in that it conducted a census of mother-baby pairs enrolled in eMTCT program in the district for a 6year period, thereby giving a comprehensive account of the successes of eMTCT option B+ in Kaberamaido district.

## Conclusion

This study has shown that there is a high rate of retention of mother-baby pairs in the eMTCT program at 12 months, while retention at 18 months is slightly lower than desired target. There was delayed conduct of 1st PCR testing as almost a third of the babies were tested late. The eMTCT program should focus on innovative strategies to enhance earlier identification of HIV positive pregnant women to improve their retention in care. There is need to promote continuous community education and sensitization on HIV/AIDS prevention, care, treatment and support with focus on enrolling HIV positive mothers early on ART to sustain the gains so far attained.

## Supporting information

**S1 File.**
(XLSX)

**S2 File.**
(DOCX)

## Acknowledgments

The authors would like to extend their appreciation to all the lecturers of Busitema University, Faculty of Health Sciences for the guidance offered during proposal development and conduct of this research. We would also like to acknowledge Mr. Ogwang Bernard, the Chief Administrative Officer Kaberamaido District Local Government, for allowing us to conduct this study in his district. Finally, we acknowledge the contribution of our HIV/AIDS service delivery

partner in the district, The AIDS Support Organization, Soroti regional project for supporting high quality HIV services in the district.

## Author Contributions

**Conceptualization:** James Daniel Odongo.

**Data curation:** James Daniel Odongo, Ronald Opito.

**Formal analysis:** James Daniel Odongo, Ronald Opito.

**Funding acquisition:** James Daniel Odongo.

**Investigation:** James Daniel Odongo.

**Methodology:** James Daniel Odongo, Benon Wanume, Denis Bwayo, Joseph K. B. Matovu.

**Project administration:** James Daniel Odongo.

**Resources:** James Daniel Odongo.

**Software:** James Daniel Odongo.

**Supervision:** Benon Wanume, Denis Bwayo, Joseph K. B. Matovu.

**Validation:** Benon Wanume, Denis Bwayo, Joseph K. B. Matovu.

**Writing – original draft:** James Daniel Odongo.

**Writing – review & editing:** James Daniel Odongo, Benon Wanume, Denis Bwayo, David Mukunya, Samuel Okware, Joseph K. B. Matovu.

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
