## [Decision Letter · Decision Letter 0]

6 Feb 2023

PONE-D-22-29453Factors associated with retention of mother-baby pairs in the elimination of mother-to-child transmission of HIV program in Kaberamaido district: a retrospective cohort study.PLOS ONE

Dear Dr. ODONGO,

Thank you for submitting your manuscript to PLOS ONE. After careful consideration, we feel that it has merit but does not fully meet PLOS ONE’s publication criteria as it currently stands. Therefore, we invite you to submit a revised version of the manuscript that addresses the points raised during the review process.

We look forward to receiving your revised manuscript.

Kind regards,

Orvalho Augusto, MD, MPH

Academic Editor

PLOS ONE

Journal Requirements:

3. We note you have included a table to which you do not refer in the text of your manuscript. Please ensure that you refer to Table 2 in your text; if accepted, production will need this reference to link the reader to the Table.

Additional Editor Comments :

Factors associated with retention of mother-bay pairs in the elimination of mother-to-child transmission of HIV program in Kaberamaido district: a retrospective cohort study

PONE-D-22-29453

This is an interesting study. The authors claim to have conducted a retrospective cohort study to 1) determine the levels of retention in care rates and 2) identify factors associated with the retention in care of mother-baby pairs along the eMTCT in a large district in Uganda. It is an example of how to use hospital-based registers to mount a follow-up study and provide long-term important outcomes.

However, there are so many issues:

1. Is it a cohort study? By definition, a cohort study requires a clear definition of exposure. Here there is no clear exposure. Yes, there is a follow-up and clear outcomes defined. Please fill out the STROBE checklist form (https://www.strobe-statement.org/checklists/) for cohort studies and annexes. By doing so the co-authors will improve the reporting.

2. The background needs some reorganization:

- The paragraph at lines 102 to 107 should come after the paragraph at lines 109 to 114.

- why citation 1 is followed by citations 10 and 11 in the first paragraph?

3. Lines 150 and 151 - the prevalence here is the overall population of adults, is itn´t? Please be specific.

4. Data abstraction procedures: please indicate the full list of variables and do not such lists with etc!

5. It is late already to correct this. The sample size formula is for just determining a proportion (typically in cross-sectional studies). It is not for looking for association analysis is done here.

6. Data analysis and Results:

- It is unclear from the description of the methods whether the authors ascertained the exact follow-up times for each child. That way it would be possible to show a Kaplan-Meier plot of retention

- A bit unusual. The follow-up time should be the age of the child because it is at 18 months old that we declare definitely negative or not. Right? Why did the authors decide to use some other follow-up time?

- The retention is estimated from your sample. Please include 95% confidence intervals. Put those as well in the abstract.

- The calendar time is ignored in this analysis. At least have the year of recruitment as one covariate to add to the model (inlcude it as dummy indicators).

- table 1:

infant age at recruitment please add more descriptives (mean, SD, range, quartiles and keep the dichotomized version as you have). Almost a third of children get to eMTCT past the recommend 8 weeks. We wonder if they do not reach 12 months (after the whole essential immunization for example). This could inform where to act to reduce these losses of opportunity.

Mothers' ART entry point the eMTCT subdivide (pre-natal, delivery, postdelivery?)

- lines 290 to 293: the average time spent by mother-baby pairs in eMTCT program. How this was computed? Does this account for the fact that there is right censoring time here?

- So the authors decided to study the association at 12 months and at 18 months of follow-up. There were only 19 events (people leaving the cohort) up to the 12th month of follow-up, and another 29 between the 12th and 18th months of follow-up. So for the 12 months, we do not see much when adjusted. Therefore, a) put the current tables 2 and 3 in the annexes. Then do one analysis using a discrete logistic survival model. In this model you have dummy indicators for 12 and 18 months and put the other variables as you have done. Do not include the viral load on the adjusted model.

- table 2:

Add a row for total pairs included

What is the definition for Viral load suppression here?

Only 232 mothers had a viral load ie adjusting for this variable you throw away 136 observations. I would recommend not to include this variable. Or do one analysis without this variable and move the current adjusted model with just 232 observations to supplementary materials.

- Figure 1: No need for the bars. Just the line is OK and please add confidence intervals.

Reviewers' comments:

Reviewer's Responses to Questions

**Comments to the Author**

1. Is the manuscript technically sound, and do the data support the conclusions?

Reviewer #1: Yes

2. Has the statistical analysis been performed appropriately and rigorously? 

Reviewer #1: I Don't Know

3. Have the authors made all data underlying the findings in their manuscript fully available?

Reviewer #1: Yes

4. Is the manuscript presented in an intelligible fashion and written in standard English?

Reviewer #1: No

5. Review Comments to the Author

Reviewer #1: I do have some comments to be addressed:

1. What does EID mean? EID (in the abstract and introduction) may even be a universal acronym but it should be explained the first time it is used (which is only found at the study design section)

2. From how many records were selected the 368 records included in the analysis.

3. There is a way to know if there are differences between the pairs included in the analysis compared to those not included. Those not included with incomplete data could be the most missing and lost-to-follow up with a certain implication on the study results.

4. The major limitation of the study is the inclusion criterion "On the other hand, clients with incomplete/missing information, clients transferred from other health facilities and clients not in the EID/ART register were excluded from the study", which may have led to a about estimation of results.

5. Improve formatting and capitalization

6. PLOS authors have the option to publish the peer review history of their article (what does this mean?). If published, this will include your full peer review and any attached files.

Reviewer #1: No

---

## [Author Response · Author response to Decision Letter 0]

8 Apr 2023

March 22, 2023

The Editor 

PLoS ONE

Dear Sir,

Re: RESPONSE TO COMMENTS RAISED ON MS#: PONE-D-22-29453

Please find enclosed our revised manuscript based on the comments from the Academic Editor and the peer-reviewers.

We are glad for the opportunity to revise the paper, which has improved clarity of the main message in the paper. We thank the reviewers for their insightfulness and the guidance from the Academic Editor.

We look forward to our paper being published in your prestigious journal. Please find below our point-by-point response to the comments, highlighted in red.

Regards,

Dr James Odongo

Corresponding Author

POINT-BY-POINT RESPONSE TO THE ACADEMIC EDITOR AND REVIEWERS’ COMMENTS

A: RSPONSE TO ACADEMIC EDITOR COMMENTS

Response: The manuscript has been edited to meet PLOS ONE’s style requirements.

2. In your Data Availability statement, you have not specified where the minimal data set underlying the results described in your manuscript can be found. 

Response: The data has been made available, marked, and uploaded online as S2 (Supporting File 2).

3. We note you have included a table to which you do not refer in the text of your manuscript. Please ensure that you refer to Table 2 in your text; if accepted, production will need this reference to link the reader to the Table. 

Response: Table 2 and 3 have been removed and moved to supplementary tables.

Additional Editor Comments:

1. Is it a cohort study? By definition, a cohort study requires a clear definition of exposure. Here there is no clear exposure. Yes, there is a follow-up and clear outcomes defined. Please fill out the STROBE checklist form (https://www.strobe-statement.org/checklists/) for cohort studies and annexes. By doing so the co-authors will improve the reporting. 

Response: This is a cohort study, and the exposure was babies being exposed to HIV infection from their mothers. We have re-aligned the work to the STROBE guidelines. A completed copy of the STROBE checklist is attached as S3 (Supplementary File 3).

2. The background needs some reorganization:

- The paragraph at lines 102 to 107 should come after the paragraph at lines 109 to 114. 

Response: This has been re-aligned. 

- why citation 1 is followed by citations 10 and 11 in the first paragraph? 

Response: Citation numbering has been corrected. 

3. Lines 150 and 151 - the prevalence here is the overall population of adults, is itn´t? Please be specific. Response: This has been clarified; it’s the overall prevalence of HIV among adults. 

4. Data abstraction procedures: please indicate the full list of variables and do not such lists with etc! Response: The full list of variables has been indicated. (see line 233 to 235 & 242 to 250)

5. It is late already to correct this. The sample size formula is for just determining a proportion (typically in cross-sectional studies). It is not for looking for association analysis is done here.

Response: We have acknowledged the limitations posed by our sample size formula but as the Editor noted, this is difficult to correct now.

6. Data analysis and Results:

- It is unclear from the description of the methods whether the authors ascertained the exact follow-up times for each child. That way it would be possible to show a Kaplan-Meier plot of retention. 

Response: Figure 1 shows the Kaplan-Meier plot of retention of the mother baby pairs and the total follow up time has also been summarized in the text.( see page 14) 

- A bit unusual. The follow-up time should be the age of the child because it is at 18 months old that we declare definitely negative or not. Right? Why did the authors decide to use some other follow-up time? 

Response: Indeed, the follow up time was measured in reference to the age of the child and censoring was done for all still active at 18months. 

- The retention is estimated from your sample. Please include 95% confidence intervals. Put those as well in the abstract. 

Response: This was included in the retention and lost to follow up estimates. 

- The calendar time is ignored in this analysis. At least have the year of recruitment as one covariate to add to the model (inlcude it as dummy indicators).

Response: The calendar time was not included in the analysis because time-series analysis focuses on the time to event (how long did the participant take to experience an outcome of interest/failure) and there is staggered entry of participant in the study. 

7. Table 1: infant age at recruitment please add more descriptives (mean, SD, range, quartiles and keep the dichotomized version as you have). 

Response: This has been included in the table description.

8. Almost a third of children get to eMTCT past the recommend 8 weeks. We wonder if they do not reach 12 months (after the whole essential immunization for example). This could inform where to act to reduce these losses of opportunity. 

Response: This is true, we have included the observation that a third of the babies receive 1st PCR test late and the need for early identification and enrollment of mother-baby pairs in the eMTCT program under conclusions and recommendations. (see line 445 to 446)

9. Mothers' ART entry point the eMTCT subdivide (pre-natal, delivery, postdelivery?). 

Response: The entry point of OPD and eMTCT was picked as per the register. However, there is another covariate for the point of ART initiation which is sub divided as pre-ANC (pre-pregnancy), ANC, and postnatal (post-delivery) period, which is included in the analysis. Delivery period could not stand alone as extremely few mothers initiated ART during delivery. 

10. Lines 290 to 293: the average time spent by mother-baby pairs in eMTCT program. How this was computed? Does this account for the fact that there is right censoring time here? 

Response: The average time spent by mother-baby pairs in the eMTCT program has been adjusted and survival data summarized as the total follow up time and the average time spent in eMTCT program by those who were lost to follow-up. All mother-baby pairs who were active in care were censured at 18 months.

11. So the authors decided to study the association at 12 months and at 18 months of follow-up. There were only 19 events (people leaving the cohort) up to the 12th month of follow-up, and another 29 between the 12th and 18th months of follow-up. So for the 12 months, we do not see much when adjusted. Therefore, a) put the current tables 2 and 3 in the annexes. Then do one analysis using a discrete logistic survival model. In this model you have dummy indicators for 12 and 18 months and put the other variables as you have done. Do not include the viral load on the adjusted model. 

Response: Discrete logistic survival model was done and results presented and labeled as table 2. (see page15). The initial table 2 and 3 have been pushed to the annexes as advised. 

12. Table 2: Add a row for total pairs included. What is the definition for Viral load suppression here? Only 232 mothers had a viral load ie adjusting for this variable you throw away 136 observations. I would recommend not to include this variable. Or do one analysis without this variable and move the current adjusted model with just 232 observations to supplementary materials. 

Response: Viral load has been excluded from the final table with adjusted hazard ratios. 

13. Figure 1: No need for the bars. Just the line is OK and please add confidence intervals. 

Response: This has been adjusted as a Kaplan-Meier curve with confidence interval included.(see page 14) 

B: RESPONSE TO REVIEWERS’ COMMENTS

Reviewer #1: 

1. What does EID mean? EID (in the abstract and introduction) may even be a universal acronym but it should be explained the first time it is used (which is only found at the study design section). 

Response: EID is early infant diagnosis. This explanation has been included in the abstract and introduction section of the manuscript (see line51).

2. From how many records were selected the 368 records included in the analysis. 

Response: This was from a sample of 457. All the mother-baby pairs with complete records were included in the analysis and only those with incomplete record excluded. (see page 8)

3. There is a way to know if there are differences between the pairs included in the analysis compared to those not included. Those not included with incomplete data could be the most missing and lost-to-follow up with a certain implication on the study results. 

Response: We acknowledge the fact that there could be a difference between the pairs included in analysis compared to those not include. This was a major limitation to our study and has been discussed under study limitations (line 419 to 421). 

4. The major limitation of the study is the inclusion criterion "On the other hand, clients with incomplete/missing information, clients transferred from other health facilities and clients not in the EID/ART register were excluded from the study", which may have led to a about estimation of results. 

Response: We acknowledge this and have included it as a major limitation to the study. (see line 419 to 421)

5. Improve formatting and capitalization. 

Response: This has been done.

---

## [Decision Letter · Decision Letter 1]

23 May 2023

PONE-D-22-29453R1Factors associated with retention of mother-baby pairs in the elimination of mother-to-child transmission of HIV program in Kaberamaido district: a retrospective cohort study.PLOS ONE

Dear Dr. ODONGO,

Thank you for submitting your manuscript to PLOS ONE. After careful consideration, we feel that it has merit but does not fully meet PLOS ONE’s publication criteria as it currently stands. Therefore, we invite you to submit a revised version of the manuscript that addresses the points raised during the review process.

We look forward to receiving your revised manuscript.

Kind regards,

Orvalho Augusto, MD, MPH

Academic Editor

PLOS ONE

Journal Requirements:

Additional Editor Comments:

This report is very relevant as it documents retention rates at 12 and 18 months in the eMTCT (elimination mother-to-child transmission) program somewhere in Uganda. Since the last version there were improvements. There are a few shortcomings:

1. Study design. Be careful with the meaning of the word cohort. Indeed, the authors might have a cohort of mother-children pairs who entered in the eMTCT program and were followed up to 12-18 months. But that alone does not mean it is a cohort (the epidemiologic analytical study) study because a comparison group is not described. Remember in epidemiologic terms an exposure means at least 2 levels (exposed or not exposed) ie. at least a control group (no exposure) must be defined. Which is not the case here. The authors stated in the response to the previous round of comments that "babies being exposed to HIV infection from their mothers" is the exposure. Where is the comparator of such a group? In fact, the analysis never produces an association between retention and "being registered in the eMTCT program". Ok?

Because of this, I suggest changing the title and removing any designation of cohort study here. You may say something like "longitudinal analysis" or "cohort of mother-babies" etc…

2. It is still a bit unclear what is the definition to enter the cohort here. Line 184 says "HIV-positive mother-baby pairs enrolled in the eMTCT program from 1st January 2013 to 31st December 2018". When is this event? At PCR sample collection?

3. About the association analysis:

- Please note Cox in "Cox proportional hazard regression" is a name. Please write with capital "C".

- Usually there are 3 time scales to deal in a cohort analysis. The age (child and mother ages), the follow up time and calendar time. The authors i) dealt with the ages (mother age is used for adjustment; child age is homogeneous so OK to ignore for now); ii) the follow up time [which the Cox PH uses, so OK]; iii) Calendar time should not be ignored. For example, the test and treat introduction in 2016 might have changed how and/or characteristics of mother-baby pairs going to eMTCT, and also could affect the retention. Therefore calendar time could be a confounder here. It would be OK to just add dummy indicator of the year the mother-baby entered in the eMTCT. Also add in the table 1 the year of when the pair entered in the eMTCT.

4. Do not only "excel" in line 255. It is "Microsoft excel" and please add a citation and version.

5. Line 247 mentions viral load. Of whom? Of the mother or of the child?

6. Figure 2.

- I apologize to the authors that I made them change the time scale to age. Please return to follow up time (in months).

- Add please add below the plot the risk set size.

7. Figure 1:

- The totals do not add up. This is because we need a box showing the 368 who entered the analysis. So between the 457 and the 368 there were 89 removed.

8. The 89 who where excluded due to incomplete data represent 19.5% of all potential mother-baby pairs. Please, add a comparison of the 89 and 457.

Reviewers' comments:

Reviewer's Responses to Questions

**Comments to the Author**

1. If the authors have adequately addressed your comments raised in a previous round of review and you feel that this manuscript is now acceptable for publication, you may indicate that here to bypass the “Comments to the Author” section, enter your conflict of interest statement in the “Confidential to Editor” section, and submit your "Accept" recommendation.

Reviewer #2: (No Response)

2. Is the manuscript technically sound, and do the data support the conclusions?

Reviewer #2: Partly

3. Has the statistical analysis been performed appropriately and rigorously? 

Reviewer #2: Yes

4. Have the authors made all data underlying the findings in their manuscript fully available?

Reviewer #2: Yes

5. Is the manuscript presented in an intelligible fashion and written in standard English?

Reviewer #2: Yes

6. Review Comments to the Author

Reviewer #2: The manuscript reports on the very important topic of elimination of mother to child transmission of HIV in Uganda. It focuses on retention of mother-baby pairs, which is critical for the success of any EMTCT program. The manuscript is generally well written; however, the authors may wish to address the following:

Abstract: Line 41-42. The authors may wish to specify that the statistics reported in this sentence are for Uganda.

Introduction: Lines 106-107. please add a reference to the transmission risk of 20-25% referred to in this sentence.

Material and Methods>Study area. Line 153. The authors indicate here that the study was carried out in 5 health facilities in Kaberamaido district. In line 157-158, they indicate that 8 health facilities are providing HIV/AIDS services in the district; and in lines 184-185, the authors state that all HIV-positive mother-baby pairs in the district were enrolled. Is the data presented in the manuscript from 5 or 8 health facilities? Can the authors clarify this in the METHODS section?

The authors should try to provide additional information about the study health facilities. For example, how many of these facilities were health centers and how many were hospitals? What routine PMTCT services are offered in these facilities. When is the first PCR? when is the second PCR?

Line 154-169: The authors have provided valuable information about Kaberamaido district. Please cite the source of this information.

Material and Methods>Study Design: Line 180. The follow-up time for the cohort is 24 months. At what point does this follow-up start? is it during pregnancy? At delivery? or some other time? Please clarify. The RESULTS indicate that follow-up ends when the baby is 18 months of age; how do the authors reconcile the 18 months with the 24 months stated here?

Materials and Methods>Sample size determination. The formula used is appropriate for a prevalence study, not this kind of study. This should specifically be mentioned in the limitations.

Materials and Methods>Data abstraction procedures. Line 221-222. The sentence here appears incomplete. Edit it.

Measurement of variables. Line 247-248; Define viral suppression.

Line 249-251. Here, the time of Nevirapine initiation is categorized as before or after 24 hours, but in the RESULTS section, Table 1, Nevirapine prophylaxis is categorized as before or after 72 hours. Be consistent in your definition and categorization of this variable.

DISCUSSION: Line 355-357. TASO-Soroti is mentioned here for the first time without defining it! What does TASO do? What support do they provide to the health facilities? The authors should describe TASO and the kind of work they do in the METHODS section. It is hard to follow the argument in the DISCUSSION without clear understanding of the work done by this organization.

Line 369-372; Health system strengthening by Implementing Partners is given as a possible explanation for the high retention. A brief description of these health system strengthening activities should be written in the METHODS section to give the reader a better understanding of the study setting and routines. To mention this late in the DISCUSSION makes it difficult for the reader to follow the explanations given.

Line 381: LTFU is mentioned here and in Table 2. Please write it in full at first mention.

Line 424-429: The statements here seem to suggest that the authors attempted to reduce misclassification of loss to follow-up by calling participants to determine true outcomes. Participants who could not be traced by phone were left out of the study because of resource limitations (line 428-429). This raises a number of questions. The study was designed to use secondary data from routine facility records, how is it that you added primary data collected through phone interviews? If the phone interviews were carried out for the purpose of collecting research data, did the participants provide any consent? Why did the authors leave out participants who could not be traced? Shouldn't they have been classified as lost to follow-up?

Line 437-439: The message in this sentence is not clear. The sentence is probably incomplete.

Table 3>Viral load status. This appears to be the only significant variable in the adjusted model. It would be important to know at what point these viral load results were obtained. Are these results of tests carried out during pregnancy, post-natal period or at any time during study follow-up? How did you define suppression in instances where some participants have 2 or more viral load results in the course of study follow-up? If a test carried out in pregnancy showed viral suppression and later in the post-natal period a subsequent test showed non-suppression or vice-versa, how were these results categorized? Can you clearly define these categories in the METHODS section.

7. PLOS authors have the option to publish the peer review history of their article (what does this mean?). If published, this will include your full peer review and any attached files.

Reviewer #2: No

---

## [Author Response · Author response to Decision Letter 1]

12 Jun 2023

Please find included in the uploaded files a point by point response to the editor's and reviewer's comments.

---

## [Decision Letter · Decision Letter 2]

29 Jun 2023

Factors associated with retention of mother-baby pairs in the elimination of mother-to-child transmission of HIV program in Kaberamaido district: a retrospective cohort study.

PONE-D-22-29453R2

Dear Dr. ODONGO,

We’re pleased to inform you that your manuscript has been judged scientifically suitable for publication and will be formally accepted for publication once it meets all outstanding technical requirements.

Kind regards,

Orvalho Augusto, MD, MPH

Academic Editor

PLOS ONE

Additional Editor Comments (optional):

This manuscript has improved. Few minor issues:

Line 70 (in the abstract) change multivariate to multivariable.

Lines 292./293 and 300 add "hazards" so the name of the model becomes "Cox proportional hazards".

Reviewers' comments:

Reviewer's Responses to Questions

**Comments to the Author**

1. If the authors have adequately addressed your comments raised in a previous round of review and you feel that this manuscript is now acceptable for publication, you may indicate that here to bypass the “Comments to the Author” section, enter your conflict of interest statement in the “Confidential to Editor” section, and submit your "Accept" recommendation.

Reviewer #2: All comments have been addressed

2. Is the manuscript technically sound, and do the data support the conclusions?

Reviewer #2: Yes

3. Has the statistical analysis been performed appropriately and rigorously? 

Reviewer #2: Yes

4. Have the authors made all data underlying the findings in their manuscript fully available?

Reviewer #2: Yes

5. Is the manuscript presented in an intelligible fashion and written in standard English?

Reviewer #2: Yes

6. Review Comments to the Author

Reviewer #2: (No Response)

7. PLOS authors have the option to publish the peer review history of their article (what does this mean?). If published, this will include your full peer review and any attached files.

Reviewer #2: No

---

## [Editor Report · Acceptance letter]

12 Jul 2023

PONE-D-22-29453R2 

Factors associated with retention of mother-baby pairs in the elimination of mother-to-child transmission of HIV program in Kaberamaido district: a longitudinal analysis. 

Dear Dr. Odongo:

I'm pleased to inform you that your manuscript has been deemed suitable for publication in PLOS ONE. Congratulations! Your manuscript is now with our production department. 

Kind regards, 

on behalf of

Dr. Orvalho Augusto 

Academic Editor

PLOS ONE